# Testability of Instrumental Variables in Linear Non-Gaussian Acyclic Causal Models

**DOI:** 10.3390/e24040512

**Published:** 2022-04-05

**Authors:** Feng Xie, Yangbo He, Zhi Geng, Zhengming Chen, Ru Hou, Kun Zhang

**Affiliations:** 1School of Mathematical Sciences, Peking University, Beijing 100871, China; xiefeng@math.pku.edu.cn (F.X.); rhou@pku.edu.cn (R.H.); 2School of Mathematics and Statistics, Beijing Technology and Business University, Beijing 100048, China; zhigeng@btbu.edu.cn; 3School of Computer, Guangdong University of Technology, Guangzhou 510006, China; 2111905124@mail2.gdut.edu.cn; 4Department of Philosophy, Carnegie Mellon University, Pittsburgh, PA 15213, USA; kunz1@cmu.edu; 5Machine Learning Department, Mohamed bin Zayed University of Artificial Intelligence, Abu Dhabi 7909, United Arab Emirates

**Keywords:** instrumental variable, causal graph, non-Gaussianity, causal discovery

## Abstract

This paper investigates the problem of selecting instrumental variables relative to a target causal influence 
X→Y
 from observational data generated by linear non-Gaussian acyclic causal models in the presence of unmeasured confounders. We propose a necessary condition for detecting variables that cannot serve as instrumental variables. Unlike many existing conditions for continuous variables, i.e., that at least two or more valid instrumental variables are present in the system, our condition is designed with a single instrumental variable. We then characterize the graphical implications of our condition in linear non-Gaussian acyclic causal models. Given that the existing graphical criteria for the instrument validity are not directly testable given observational data, we further show whether and how such graphical criteria can be checked by exploiting our condition. Finally, we develop a method to select the set of candidate instrumental variables given observational data. Experimental results on both synthetic and real-world data show the effectiveness of the proposed method.

## 1. Introduction

Estimating causal effects from observational data is an important problem, especially in the presence of unmeasured confounding. The instrumental variable (IV or instrument) model is a general approach to estimate causal effect in the presence of unobserved variables [1,2,3,4] and is used in a wide range of literature, such as economics [5,6], sociology [4,7], and epidemiology [8,9].

A major challenging problem in an instrumental variable model is how to select a valid IV to infer the causal effect of one variable *X* on another variable *Y*. In general, IVs need to be chosen based on domain knowledge or expert experience. However, it is sometimes difficult to select a valid IV without precise prior knowledge of causal structure, and an invalid IV may cause a biased estimation of the effect of *X* on *Y* [10]. Therefore, it is desirable to investigate ways of selecting IVs only from observed variables.

Although it is not possible to test whether a variable is a valid IV only from the joint distribution of observed variables, there exist several methods for testing whether a variable of interest is an invalid IV. Pearl [11] provided a necessary condition, called the *instrumental inequality*,for a general instrument model, which can be used to test whether a variable is a candidate IV for discrete variables. Inspired by instrumental inequality, various contributions were made towards discovering the testability of IV validity in different scenarios [12,13,14,15]. More recently, Kédagni and Mourifié [16] considered a more general case where treatment is discrete and there are no restrictions on IV and outcome and proposed generalized instrumental inequalities to test the IV independence assumption. However, those approaches fail to work when treatment is a continuous variable. Pearl [11] conjectured that instrument validity cannot be tested in the case where treatment is a continuous variable without any further assumption, which was recently proved by Gunsilius [17].

There exist works in the literature that address the continuous variable setting. Kuroki and Cai [18] utilized vanishing Tetrad conditions [19] and proposed a new necessary condition to solve this problem in the linear structural causal model. However, their method needs at least three valid IVs in the observed variables. Kang et al. [20] proposed the sisVIVE algorithm to estimate the causal effect in the case where more than half of the variables are valid IVs in the observed variables. Later, Silva and Shimizu [21] appear to be the first to exploit the non-Gaussianity property in the linear structural causal model. They utilized the generalized Tetrad conditions (t-separation) [22,23] and designed a IV-TETRAD algorithm to select IVs. Unfortunately, their conditions still require two or more IVs as a prerequisite for instrument testing and may rule out some correct IVs. For instance, consider the causal graph in Figure 1. Assume the causal relationships between variables are linear and that the noise terms follow non-Gaussian distributions. Then, the IV-TETRAD returns an empty set of candidate IVs though *Z* is a valid IV relative to 
X→Y
.

In this paper, we show that, for continuous data, a single variable *Z* being a valid IV relative to 
X→Y
 imposes certain constraints in a linear non-Gaussian acyclic causal model. Specifically, we make the following contributions:1.We propose a necessary condition for detecting variables that cannot serve as (conditional) IVs by the so-called generalized independent noise (GIN) condition [24], which is called instrumental variable generalized independent noise (IV-GIN) condition. We characterize the graphical implications of IV-GIN condition in linear non-Gaussian acyclic causal models.2.We then further show whether and how the graphical criteria of an instrumental variable can be checked by exploiting the IV-GIN conditions.3.We develop a method to select the set of candidate IVs for the target causal influence 
X→Y
 from the observational data by IV-GIN conditions.4.We demonstrate the efficacy of our algorithm on both synthetic and real-word data.

## 2. Related Work

In this section, we review some of the key works that are most closely related to ours.

### 2.1. Instrument Variable Models

The instrumental variable (IV) model is a general approach to estimate the causal effect of a treatment *X* on an outcome *Y* of interest in presence of unobserved variables [1,2,3]. That is to say, the IV model is an unbiased estimator of the causal effect of *X* on *Y* of interest [4,6]. In practice, one can obtain IVs based on domain knowledge or expert experience. However, it is sometimes difficult to select the valid IV without precise prior knowledge of causal structure, and an invalid IV may cause a biased estimation of the effect of *X* on *Y* [10]. In this paper, we investigate data-driven ways of selecting IVs only from observed variables. The current methods for selecting IVs can be roughly divided into the following two settings.

In the literature of the discrete variable setting, Pearl [11] provided a necessary condition, called *instrumental inequality*, which can be used to test whether a variable is an invalid IV. Inspired by instrumental inequality, various contributions were made to discover IV validity’s testability in different scenarios. For instance, Manski [12] showed the same instrumental inequality in the missing data model. Palmer et al. [13] and Wang et al. [15] considered useful tests of the instrumental inequality in the binary instrumental variable model. Kitagawa [14] introduced another test of the instrument in the case where the outcome is continuous. More recently, Kédagni and Mourifié [16] proposed generalized instrumental inequalities to test the IV independence assumption in the case where treatment is discrete and there are no restrictions on IV and outcome. Gunsilius [17] recently proved the Pearl’s conjecture that instrument validity cannot be tested in the case where treatment is a continuous variable without any further assumption [11].

There exist works in the literature that address the continuous variable setting. For instance, Kuroki and Cai [18] proposed a new necessary condition to resolve this problem in the linear structural causal model using the so-called Tetrad conditions [19]. Later, Kang et al. [20] proposed the sisVIVE algorithm to estimate the causal effect in the case where more than half of the candidate instruments are valid (*majority rule*). Recently, Silva and Shimizu [21] appear to be the first to exploit the non-Gaussianity property in the linear structural causal model. They designed an IV-TETRAD algorithm to select IVs using the generalized Tetrad conditions (t-separation) [22,23]. Unfortunately, the above methods require two or more IVs as a prerequisite for instrument testing, and some methods (e.g., IV-TETRAD approach) may rule out some correct IVs.

Our work focuses on the continuous setting. Unlike the existing works, we show that a single variable *Z*, being a valid IV relative to 
X→Y
, imposes certain constraints in a linear non-Gaussian acyclic causal model.

### 2.2. Causal Graphical Models

Graphical models with latent variables are extensively studied in the literature. Unlike the existing methods of learning the undirected graphical model [25,26,27,28,29,30,31,32,33], here, we focus only on the most closely related work on causal graphical models, i.e., a directed acyclic graph (DAG) *G* representing the relations of causation among the variables [4,7]. Within the space of discovering a causal graphical model on observed data, the commonly used strategies are as follows.

One typical strategy for handling this problem is using conditional independence tests to learn the causal graph over the observed variables [4,7]. Well-known algorithms along this line include Fast Causal Inference (FCI) [34], Really Fast Causal Inference (RFCI)  [35], and their variants [36]. These methods learn the equivalence class of maximal ancestral graphs (MAGs), as represented by PAG (partial ancestral graph). However, these works focus on estimating the causal structure over only observed variables and can not recover the precise causal graph. In our work, we try to discover the set of candidate IVs from observational variables without prior knowledge of causal graphs.

Another strategy is functional causal model-based approaches. For instance, Hoyer et al. [37] showed that the causal order between any two observed variables is identifiable in the linear non-Gaussian causal model. Later, more efficient methods were proposed to learn the causal graph over observed variables  [38,39]. Recently, Salehkaleybar et al. [40] showed that the set of all possible causal effects between any two observed variables is identifiable in the same setting. Unfortunately, the size of the equivalence class of the identified causal effects could be very large, and their method requires specifying the number of latent variables a priori [21].

There is also an interesting strategy based on the “Sparse plus Low Rank Matrix Decomposition”. Many methods are proposed to address the challenge of learning a latent Gaussian graph model. For instance, Chandrasekaran et al. [26] formulated a convex objective involving nuclear norm penalization maximum likelihood for Gaussian graphical model estimation with a few latent confounders. Zorzi and Sepulchre [28] presented a two-step procedure for estimating autoregressive (AR) latent variable graphical models. Later, Ciccone et al. [41] reformulated this decomposition problem for the setting where only the sample covariance is available, and the difference between the sample covariance and the actual one is non-negligible. Alpago et al. [42] proposed an identification procedure for a sparse graphical model associated with a reciprocal process. However, these methods focus on the undirected graphical model. In the field of a causal graphical model, Frot et al. [43] introduced the LRpSC+GES algorithm to learn the causal structure with some hidden variables. Agrawal et al. [44] proposed a practical algorithm, the DeCAMFounder, to consistently estimate causal relationships in the nonlinear, pervasive confounding setting. Although these methods are used in a range of fields, they usually assume that the underlying graph among the observed variables is sparse, and there are a few hidden variables that have a direct effect on many of the observed variables. The modeling of our paper does not restrict those assumptions and allows arbitrary hidden structures.

In summary, unlike the existing methods of recovering causal graphical models, our goal is to select the set of candidate IVs from observational variables without precise prior knowledge of causal graph.

## 3. Preliminaries

### 3.1. Notation and Graph Terminology

We follow the notational conventions used in  [7]. Let *G* be a directed acyclic graph (DAG) with the nodes (or vertex) set 
V
 and the directed edges set 
E
. Here, we use “variable” and “node” interchangeably. A **path** is a sequence of nodes 
$\left\{V_{1},\ldots ,V_{r}\right\}$
 such that 
$V_{i}$
 and 
$V_{i+1}$
 are adjacent in *G*, where 
1≤i<r
. Furthermore, if the edge between 
$V_{i}$
 and 
$V_{i+1}$
 has its arrow pointing to 
$V_{i+1}$
 for 
i=1,2,…,r−1
, we say that the path is **directed** from 
$V_{1}$
 to 
$V_{r}$
. A **collider** on a path 
$\left\{V_{1},\ldots ,V_{p}\right\}$
 is a node 
$V_{i}$
, 
1<i<p
, such that 
$V_{i-1}$
 and 
$V_{i+1}$
 are parents of 
$V_{i}$
. We say a path is **active** if this path can be traced without traversing a collider. A **trek** between 
$V_{i}$
 and 
$V_{j}$
 is a path that does not contain any colliders in *G*. The set of all parents and children of 
$V_{i}$
 are denoted by 
$\mathrm{Pa}(V_{i})$
 and 
$\mathrm{Ch}(V_{i})$
, respectively. Besides, for a set 
O
, 
|O|
 denotes the number of elements of set 
O
. Other commonly used concepts in graphical models, such as d-separation, can be found in  [4,7].

### 3.2. Instrumental Variable Model

Here, we follow the notational conventions and definitions used in [45]. Let *X* be the treatment (exposure), *Y* be the outcome, and 
U
 be the set of unmeasured confounders between *X* and *Y*.

**Definition** **1**((Conditional) Instrumental Variable Criteria). *Given the causal graph G, a variable Z is a (conditional) instrumental variable to a target causal effect 
X→Y
 given 
W
, if and only if it satisfies the following conditions:*
*1.* *
W
 contains only nondescendants of Y in G;**2.* *
W
 d-separates Z from Y in the graph obtained by removing the edge 
X→Y
 from G;**3.* *
W
 does not d-separates Z from X in G.*

For simplicity, we call these three conditions *instrument criteria*.

**Definition** **2**(IV Estimator). *Suppose variable Z is a (conditional) IV for 
X→Y
 given 
W
, the causal effect of X on Y, denoted by 
$b_{YX}$
, is identified in a linear model and given by*

(1)
$$\begin{array}{c} b_{YX}=\frac{\sigma_{ZY\cdot W}}{\sigma_{ZX\cdot W}}, \end{array}$$

*where 
$\sigma_{ZY\cdot W}$
 denotes the partial covariance between Z and Y given the set 
W
, and 
$\sigma_{ZX\cdot W}$
 denotes the partial covariance between Z and X given the set 
W
.*

Figure 2 illustrates a simple instrumental variable model, where *Z* is an IV conditioning on 
$\left\{W_{1},W_{2}\right\}$
 for the relation 
X→Y
. The causal effect 
$b_{YX}$
 is 
$\frac{\sigma_{ZY\cdot \{W_{1},W_{2}\}}}{\sigma_{ZX\cdot \{W_{1},W_{2}\}}}$
.

### 3.3. Problem Setup

In this paper, we assume that the system of interest is a linear non-Gaussian acyclic causal model with variables in 
V={X,Y}∪U∪O
, where *X* is the treatment, *Y* is the outcome, 
U
 is the set of unmeasured (latent or hidden) variables, and 
O
 is the set of other measured variables. In particular, without loss of generality, we assume that all variables in 
V
 have a zero mean. Each variable 
$V_{i}\in V$
 is generated according to the following linear structural equation model (SEM):
(2)
$$\begin{array}{cc} V_{i} & =\sum_{V_{j}\in \mathrm{Pa}\left(V_{i}\right)}b_{ij}V_{j}+\varepsilon_{V_{i}} \end{array}$$

where 
$b_{ij}$
 is the causal strength from 
$V_{j}$
 to 
$V_{i}$
. All noise terms 
$\varepsilon_{V_{i}}$
 are continuous random variables following non-Gaussian distributions with nonzero variances and are independent of each other. We restrict our attention to the recursive model [46]. That is to say, the causal relationships among variables can be represented by a DAG [4,7]. This model is also known as linear, non-Gaussian, acyclic model (LiNGAM) when all variables in 
V
 are observed [47].

Our problem of interest is to study the testability of IV validity for the relation 
X→Y
 in a linear non-Gaussian acyclic causal model. To this end, theoretically, we need to investigate the testability of instrument criteria from observational variables.

## 4. Necessary Condition for Instrumental Variable

In this section, we first give a simple example to show that a valid IV imposes some constraints with the help of non-Gaussianity. Then, we give our necessary condition for (conditional) IVs by using generalized independent noise (GIN) conditions [24]. Finally, we present the graphical implications of the proposed condition in linear non-Gaussian causal models. To improve readability, we defer all proofs to the Appendix A.

### 4.1. A Motivating Example

Before showing the theoretical results, let us look at two simple graphs shown in Figure 3. Suppose the generating mechanisms of two subgraphs are as follows:Subgraph (a): 
$U_{1}=\varepsilon_{U_{1}}$
, 
$Z=\varepsilon_{Z}$
, 
X=2Z+0.5U1+εX
, and 
$Y=1X+2U_{1}+\varepsilon_{Y}$
;Subgraph (b): 
$U_{1}=\varepsilon_{U_{1}}$
, 
$Z=1U_{1}+\varepsilon_{Z}$
, 
X=2Z+0.5U1+εX
, and 
$Y=1X+2U_{1}+\varepsilon_{Y}$
.

Here, we consider two cases, namely Gaussian and uniform cases:*Gaussian Case*: All noise terms in subgraphs (a) and (b) are generated from the standard Gaussian distributions.*Uniform Case*: All noise terms in subgraphs (a) and (b) are generated from the uniform distributions over the interval 
[0,1]
.

Let 
$Y-\frac{\sigma_{YZ}}{\sigma_{XZ}}X$
 be the surrogate-variable of 
{Y,X}
 relative to *Z*. Figure 4 shows the scatter plots of *Z* and 
$Y-\frac{\sigma_{YZ}}{\sigma_{XZ}}X$
 for two cases. Interestingly, in the Gaussian case, we find that no matter whether *Z* is an IV or not, *Z* and 
$Y-\frac{\sigma_{YZ}}{\sigma_{XZ}}X$
 are statistically independent, while in the uniform case, *Z* and 
$Y-\frac{\sigma_{YZ}}{\sigma_{XZ}}X$
 are statistically dependent if *Z* is an invalid IV. These observations imply that the non-Gaussianity (as indicated by the uniform distribution) is beneficial to find out whether a continuous variable is a candidate IV relative to 
X→Y
.

### 4.2. IV-GIN Condition for Instrumental Variable

Below, we give mathematical characterizations of the above observation by using the GIN condition. Before that, we first review the GIN condition formulated by  Xie et al. [24] and the Darmois–Skitovitch theorem that characterizes the independence of two linear statistics given in [48].

**Definition** **3**(GIN condition). *Let 
P
 and 
Q
 be two observed random vectors. Suppose the variables follow the linear non-Gaussian acyclic causal model. Define the surrogate-variable of 
P
 relative to 
Q
 as 
$E_{P||Q}\omega^{⊺}P$
, where ω satisfies 
$\omega^{⊺}E\left[PQ^{⊺}\right]=0$
 and 
ω≠0
. We say that 
(Q,P)
 follows the GIN condition if and only if 
$E_{P||Q}$
 is statistically independent from 
Q
.*

**Theorem** **1** **(Darmois–Skitovitch Theorem).***Define two random variables 
$V_{1}$
 and 
$V_{2}$
 as linear combinations of independent random variables 
$n_{1},\ldots ,n_{p}$
:*

(3)
$$\begin{array}{c} V_{1}=\sum_{i=1}^{p}\alpha_{i}n_{i},\;\;V_{2}=\sum_{i=1}^{q}\beta_{i}n_{i}, \end{array}$$

*where the 
$\alpha_{i},\beta_{i}$
 are constant coefficients. If 
$V_{1}$
 and 
$V_{2}$
 are independent, then the random variables 
$n_{j}$
 for which 
$\alpha_{j}\beta_{j}\neq 0$
 are Gaussian.*

The above theorem states that if there exists a non-Gaussian 
$n_{j}$
 for which 
$\alpha_{j}\beta_{j}\neq 0$
, 
$V_{1}$
 and 
$V_{2}$
 are dependent.

We now give the necessary condition of valid IVs by using GIN conditions.

**Theorem** **2****(Necessary Condition for IV).**
*Let G be a linear non-Gaussian acyclic causal model. Let treatment X, outcome Y, Z, and 
W
 be correlated random variables in G. Assume faithfulness holds. If Z is a valid IV conditioning on 
W
 relative to 
X→Y
 in G, then 
({Z,W},{X,Y,W})
 follows the GIN condition.*

We term this necessary condition the *IV-GIN (instrumental variable-generalized independent noise)* condition. For the rest of the paper, we say that 
[Z||W]
 follows the IV-GIN condition relative to 
X→Y
 if and only if 
({Z,W},{X,Y,W})
 follows the GIN condition. Theorem 2 indicates that one may test whether a variable *Z* is an invalid IV conditioning on 
W
 relative to 
X→Y
 by just testing the IV-GIN condition.

**Example** **1**(Motivating example, continued). *Let us continue to consider the two causal graphs in Figure 3. Assume that all noise terms follow non-Gaussian distributions. According to the linear generating mechanism and IV-GIN condition, for subgraph (a),*

(4)
$$Z=\varepsilon_{Z}$$


(5)
$$E_{\{Y,X\}||Z}=Y-\frac{\sigma_{YZ}}{\sigma_{XZ}}X=2U_{1}+\varepsilon_{y}.$$

*We find that there is no common non-Gaussian independent component shared by 
$E_{\{Y,X\}||Z}$
 and Z. Thus, we have 
$E_{\{Y,X\}||Z}$
 as independent from Z due to the Darmois–Skitovitch Theorem.**However, for subgraph (b),*

(6)
$$Z=\varepsilon_{U_{1}}+\varepsilon_{Z}$$


(7)
$$\begin{array}{cc} E_{\{Y,X\}||Z} & =Y-\frac{\sigma_{YZ}}{\sigma_{XZ}}X\\ & =\left(2-2.5t\right)U_{1}+\varepsilon_{y}-2t\varepsilon_{Z}-t\varepsilon_{X}, \end{array}$$

*where 
$t=\frac{2\mathrm{Var}(\varepsilon_{U_{1}})}{2.5\mathrm{Var}\left(\varepsilon_{U_{1}}\right)+2\mathrm{Var}\left(\varepsilon_{Z}\right)}$
. We find that there is one common, non-Gaussian independent component shared by 
$E_{\{Y,X\}||Z}$
 and Z, i.e., 
$\varepsilon_{Z}$
 because 
2t≠0
. Thus, we have 
$E_{\{Y,X\}||Z}$
 and Z as dependent due to the Darmois–Skitovitch theorem. These facts theoretically verify the results shown in Figure 4.*

### 4.3. Graphical Implications of IV-GIN Condition in Linear non-Gaussian causal Models

In this section, we characterize the graphical implications of the IV-GIN condition in linear non-Gaussian causal models. The following theorem shows the connection between IV-GIN condition and the graphical properties of the variables, and an illustrative example is given accordingly.

**Theorem** **3.***Suppose all variables 
V
 follow the linear non-Gaussian acyclic causal model and that faithfulness holds. Let treatment X, outcome Y, Z, and 
W
 be correlated random variables in 
V
. Then, 
[Z||W]
 follows the IV-GIN condition relative to 
X→Y
 and there is no proper subset 
$\tilde{W}$
 of 
W
 such that 
$\left[Z||\tilde{W}\right]$
 follows the IV-GIN condition relative to 
X→Y
 if and only if the following three conditions hold:*
*1.* *There exists a node 
C∈V
, 
C∉W
, such that for every trek π between a node 
$V_{p}\in \left\{X,Y,W\right\}$
 and a node 
$V_{q}\in \left\{Z,W\right\}$
, (a) π goes through at least one node in 
{C,W}
, denoted by 
$V_{k}$
, and (b) 
$V_{k}$
 has its arrow pointing to 
$V_{p}$
 in π. (In other words, 
$V_{k}$
 is causally earlier (according to the causal order) than 
$V_{p}$
 on π.)**2.* *There is at least one directed path between any one node in 
{C,W}
 and any one node in 
{X,Y}
.**3.* *There is no proper subset 
$\tilde{W}$
 of 
W
 to satisfy conditions 1 and 2.*

**Example** **2.**
*Consider the causal graphs shown in Figure 3 again. For subgraph (a), there exists a node X, and 
W=∅
 such that (1) every trek between Z and 
{X,Y}
, e.g., 
Z→X→Y
, goes through X and that (2) X has its arrow pointing to Y. Besides, there is at least one directed path between X and any one node in 
{X,Y}
. According to Theorem 3, we know that 
[Z||∅]
 follows the IV-GIN condition relative to 
X→Y
 in subgraph (a). However, for subgraph (b), we can not find a node C such that every trek between 
{Z}
 and a node in 
{X,Y}
 goes through C and C is causally earlier than 
{X,Y}
, e.g., treks 
Z→X
 and 
$Z\leftarrow U_{1}\to Y$
. This implies that 
[Z||∅]
 violates the IV-GIN condition in subgraph (b) according to Theorem 3.*


## 5. Testability of Instrument Criteria Validity in Terms of IV-GIN Conditions

In this section, we investigate the testability of instrument criteria by exploiting our IV-GIN condition. Note that the last condition of instrument criteria, i.e., that 
W
 does not d-separate *Z* from *X* in *G*, can be easily checked by the d-separation criterion because 
W
, *Z*, and *X* are observed variables [4]. Therefore, we focus next on the first two conditions of instrument criteria.

### 5.1. Condition 1 of Instrument Criteria

Below, we first show that the first condition, i.e., that 
W
 contains only nondescendants of *Y* in *G*, is testable by using IV-GIN conditions.

**Proposition** **1.**
*Let G be a linear non-Gaussian acyclic causal model. Let treatment X, outcome Y, Z, and 
W
 be correlated random variables in G. Assume faithfulness holds, conditions 
2∼3
 of instrument criteria hold, and there is no proper subset 
$\tilde{W}$
 of 
W
 such that 
$\left[Z||\tilde{W}\right]$
 follows the IV-GIN condition. If 
{Z,W}
 contains at least one descendant of Y in G, then 
[Z||W]
 must violate the IV-GIN condition.*


Proposition 1 ensures that the IV-GIN condition rules out the invalid IVs that do not satisfy condition 1 of instrument criteria, and an illustrative example is given in Example 3.

**Example** **3.**
*Let us consider the causal graph in Figure 5. We find that 
$\left[Z||W_{1}\right]$
 follows the IV-GIN condition because Z is a valid IV conditioning on 
$W_{1}$
. However, we find that 
$\left[Z||W_{2}\right]$
 violates the IV-GIN condition because 
$W_{2}$
 is the descendant of Y.*


### 5.2. Condition 2 of Instrument Criteria

Now, we study the second condition, i.e., that 
W
 d-separates *Z* from *Y* in the graph obtained by removing the edge 
X→Y
 from *G*. Given the conditional set 
W
, the condition 2 can be phrased as follows:2a.There is no active nondirected path between *Z* and *Y* that does not include *X*;2b.There is no active directed path from *Z* to *Y* that does not include *X*.

In the remainder of this subsection, we discuss these two subconditions separately.

#### 5.2.1. Subcondition 2a

It was shown that one can verify the validity of condition 2a in the case where at least two IVs are present in the ground-truth graph [21]. However, their condition is too restricted and rules out some valid IVs. (A similar conclusion is reported in Proposition 17 of [21].) Figure 1 shows an example that their method outputs an empty set of candidate IVs, though *Z* is a valid IV. In contrast, our IV-GIN condition is relatively mild and is able to avoid ruling out the valid IVs. Although one might not fully verify the validity of condition 2a using the IV-GIN condition, most invalid IVs that do not satisfy condition 2a are ruled out, as shown in the following theorem.

**Proposition** **2.**
*Let G be a linear non-Gaussian acyclic causal model. Let treatment X, outcome Y, Z, and 
W
 be correlated random variables in G. Assume faithfulness holds, conditions 1 and 3 of instrument criteria hold, and there is no proper subset 
$\tilde{W}$
 of 
W
 such that 
$\left[Z||\tilde{W}\right]$
 follows the IV-GIN condition. Furthermore, given 
W
, assume there is at least one active nondirected path between Z and Y that does not include X. If given 
W
, there is no node 
C∈V
 such that all active paths between Z and Y go through C and C has its arrow pointing to Y, then 
[Z||W]
 must violate the IV-GIN condition.*


Below, we give an example to illustrate Proposition 2.

**Example** **4.**
*Consider the causal diagram shown in Figure 6. Given 
$W_{1}$
, there is one active nondirected path between Z and Y, i.e., 
$Z\leftarrow U_{2}\to Y$
, and all active paths between Z and Y are 
Z→X→Y
, and 
$Z\to U_{2}\to Y$
. Thus, we can not find a node C such that all active paths between Z and Y go through C, and C has its arrow pointing to Y. This fact implies that 
$\left[Z||W_{1}\right]$
 violates the IV-GIN condition. That is to say, Z is an invalid IV conditioning on 
$W_{1}$
 relative to 
X→Y
.*


Now, we give a simple example to show that though the IV-GIN condition holds, the condition 2a of instrument criteria is violated.

**Example** **5.**
*Consider the causal diagram shown in Figure 7. We can find a node 
$U_{2}$
 such that all active paths between Z and Y go through 
$U_{2}$
 and 
$U_{2}$
 has its arrow pointing to Y. This implies that 
[Z||∅]
 follows the IV-GIN condition according to Proposition 2. This example tells us the IV-GIN condition is necessary, but not sufficient, to test condition 2a.*


#### 5.2.2. Subcondition 2b

We now show that it is hard to verify the validity of condition 2b, even under the non-Gaussian assumption, through the following simple example.

Let us look at the following graph in Figure 8, where *Z* is a invalid IV conditioning on an empty set relative to 
X→Y
.

Suppose the generating mechanism of the graph is as follows:
(8)
$$U_{1}=\varepsilon_{U_{1}},Z=\varepsilon_{Z},$$


(9)
$$X=\alpha Z+\gamma U_{1}+\varepsilon_{X}$$


(10)
$$Y=\beta X+\delta U_{1}+\lambda Z+\varepsilon_{Y}$$


According to the definition of GIN condition, we have

(11)
$$E_{\{Y,X\}||Z}=Y-\frac{\sigma_{YZ}}{\sigma_{XZ}}X$$


(12)
$$\;\;\;\;\;\;\;\;\;\;\;=\left(\delta -\lambda /\alpha \right)U_{1}-\left(\lambda /\alpha \right)\varepsilon_{x}+\varepsilon_{Y}),$$
 Based on the above equation, the component of 
$\varepsilon_{Z}$
 is successfully removed from 
$E_{\{Y,X\}||Z}$
 although *Y* is generated by 
$\left\{Z,X,U_{1}\right\}$
. This implies that 
$E_{\{Y,X\}||Z}$
 is independent from *Z* according to the Darmois–Skitovitch theorem. That is to say, 
$\left[Z||W_{1}\right]$
 follows the IV-GIN condition whatever the value of 
λ
 (note that there is no directed edge between *Z* and *Y* when 
λ=0
).

## 6. Algorithm for Selecting the Candidate IVs

In this section, we leverage the above results and propose a sequential algorithm to select the set of candidate IVs for the target relationship 
X→Y
 without prior knowledge of the causal structure. Notice that the validity of a variable as an IV is dependent on which set 
W
 we condition on. To identify candidate IV efficiently, given an observed variable 
$Z_{i}$
, we start with finding IV with an empty conditional set and then increase the number of conditional variables until the IV-GIN condition is satisfied or the length of conditional set equals 
|O|−1
 (Lines 2∼14 of Algorithm 1). The details of the above process are given in Algorithm 1.
**Algorithm 1:** IV-GIN **Input:** Treatment *X*, outcome *Y*, and set of observed variables 
O
. **Output:** Set of candidate 
C
 and its corresponding conditional set 
Conset
.  1: Initialize the set of candidate IVs: 
C=∅
, the conditional set: 
Conset=∅
, the length of conditional set: 
ConsetLen=0
, and 
Tag=O
;  2: **while**

ConsetLen<|Tag|

**do**  3:    **for** each variable 
$Z_{i}\in C$
 **do**  4:     **repeat**  5:         Select a subset 
W
 from 
$O\setminus Z_{i}$
 such that 
W=ConsetLen
;  6:         **if** 
[Z||W]
 follows the IV-GIN condition **then**  7:           Add 
$Z_{i}$
 into 
C
, and delete 
$Z_{i}$
 from 
Tag
;  8:           Set 
$\mathrm{Conset}(Z_{i})=W$
;  9:           Break the repeat loop of line 4;  10:         **end if**  11:      **until** all subsets with length 
ConsetLen
 in 
$O\backslash Z_{i}$
 are selected;  12:    **end for**  13:    
ConsetLen=ConsetLen+1
;  14: **end while**  15: **Return:**

C
 and 
Conset



In practice, the main issue is how to test IV-GIN conditions, i.e., for any two sets of variables 
P
 and 
Q
, we need to test the independence between 
$E_{P||Q}$
 and 
Q
. To do so, we check for pairwise independence with Fisher’s method [49] instead of testing for the independence between 
$E_{P||Q}$
 and 
Q
 directly. In particular, denote by 
$p_{k}$
, with 
k=1,2,…,|Q|
, all resulting *p*-values from pairwise independence between variables use the Hilbert–Schmidt independence criterion (HSIC)-based independence tests [50] due to the non-Gaussianity of the data. We compute the test statistic as 
$-2\sum_{k=1}^{\left|Q\right|}\log p_{k}$
, which follows the chi-square distribution with 
2|Q|
 degrees of freedom when all the pairs are independent.

**Theorem** **4**(Completeness of IV-GIN). *Suppose that the data 
V={X,Y}∪U∪O
 strictly follows the linear non-Gaussian acyclic causal model, that is, all the model assumptions are met, and the sample size is infinite. Furthermore, assume that there exists at least one valid IV Z conditioning on 
W
 for the relation 
X→Y
, where 
Z∪W⊂V
. Then, the output 
C
 of IV-GIN method must contain all valid IVs.*

## 7. Experiments on Synthetic Data

In this section, we evaluate the IV selection performance on synthetic data and demonstrate the correctness of proposed theories.

***Comparisons:*** We make comparisons with two state-of-the-art methods: the sisVIVE algorithm [20] that needs more than half of the variables to be valid IVs, and the IV-TETRAD algorithm [21] that needs two or more variables to be valid IVs. (Here, we adopt the two functions, TestTetrad and TestResiuals, to select IVs in the IV-TETRAD algorithm.) The source codes of sisVIVE and IV-TETRAD are available from https://mirrors.sjtug.sjtu.edu.cn/cran/web/packages/sisVIVE/index.html (accessed on 20 January =2022) and http://www.homepages.ucl.ac.uk/~ucgtrbd/code/iv_discovery/ (accessed on 20 January 2022), respectively.

***Scenarios:*** We designed three scenarios, as shown in Figure 9, where *X* is treatment, *Y* is outcome, the variables 
$U_{i}$
 (
i=1,2
) are unobserved, and 
$Z_{j}$
 (
j=1,…,4
) are potential IVs. For scenarios 
$S_{1}$
 and 
$S_{2}$
, nodes 
$Z_{2}$
 and 
$Z_{3}$
 both are valid IVs conditioning on an empty set relative to 
X→Y
, and node 
$Z_{1}$
 is an invalid IV due to the path 
$Z_{1}\leftarrow U_{1}\to Y$
. The key difference between scenarios 
$S_{1}$
 and 
$S_{2}$
 is that there is an active nondirected path between 
$Z_{3}$
 and *X* in 
$S_{2}$
 while not in 
$S_{1}$
. For scenario 
$S_{3}$
, 
$Z_{1}$
 is a valid IV conditioning on 
$Z_{3}$
 relative to 
X→Y
, 
$Z_{2}$
 is a valid IV conditioning on an empty set relative to 
X→Y
, 
$Z_{3}$
 is an invalid IV due to the paths 
$Z_{3}\to Y$
 and 
$Z_{3}\leftarrow U_{1}\to Y$
, and 
$Z_{4}$
 is an invalid IV due to the path 
$X\to Z_{4}\leftarrow Y$
.

***Metrics:*** To evaluate the accuracy of the selected IVs, we used the following two metrics:*Correct-selecting rate*: The number of correctly selected valid IVs divided by the total number of valid IVs in the ground-truth graph.*Selection commission*: The number of falsely detected IVs divided by the total number of selected IVs in the output 
C
 of the current algorithm.

***Experimental setup:*** We generated data by a linear non-Gaussian causal acyclic model according to the above three scenarios. In detail, the causal strength 
$b_{ij}$
 was generated uniformly in 
[−2,−0.5]∪[0.5,2]
 and the non-Gaussian noise terms were generated from exponential distributions to the second power. Here, we conducted experiments with the following tasks:T1.*Sensitivity on the effect of sample size*. We considered different sample sizes 
N=1k,3k,5k
, where *k* = 1000.T2.*Sensitivity on the effect of unmeasured confounders between X and Y*. The coefficients between 
{X,Y}
 and 
$U_{1}$
 are set such that 
$b_{XU_{1}}=b_{YU_{1}}=\lambda$
, at two levels, 
(0.125,0.25)
, as that in [21]. The sample size *N* is 5000.

We used HSIC-based independence tests [50] for the IV-GIN condition due to the non-Gaussianity of the data. Each experiment was repeated 50 times with randomly generated data, and the results were averaged.

***Results on Task T1:*** The experimental results are reported in Table 1. From the table, we can see that our proposed IV-GIN outperforms other methods with both evaluation metrics in all there scenarios and in all sample sizes, indicating that our IV-GIN condition’s testability is wider than other algorithms’ in the linear non-Gaussian causal models. We found that the IV-TETRAD algorithm does not perform well, especially in scenarios 
$S_{2}$
 and 
$S_{3}$
, indicating that it is not capable when there is an active nondirected path between valid IV and treatment *X* (scenario 
$S_{2}$
) and a single IV is present (scenario 
$S_{3}$
). We further noticed that the sisVIVE algorithm does not perform well in scenario 
$S_{3}$
. This is because fewer than half of the variables are valid IV conditioning on the same set in scenario 
$S_{3}$
.

***Results on Task T2:*** The experimental results are reported in Table 2. It is worth noting that stronger confounding makes it more difficult to select valid IVs. From the table, we found IV-GIN gives better performances than other methods with different confounding coefficients in almost all scenarios, indicating that our IV-GIN condition is more efficient than other algorithms. We noticed that although the Correct-selecting rate of sisVIVE is higher than IV-GIN in scenario 
$S_{1}$
 when 
λ=0.25
, the selection commission of IV-GIN is lower than sisVIVE (lower is better for selection commission).

To conclude, these above findings show a clear advantage of our method over the compared algorithms.

## 8. Application to Vitamin D Data

In this section, we apply our algorithm to the Vitamin D data set described by Skaaby et al. [51], where the data we analyze are the population-based study Monica10. The data we use are collected from 2571 individuals between 40–71 years, as reported in [52]. In detail, these data contain 5 variables, including treatment *Vitamin D status* (continuous variable), outcome *mortality*, *filaggrin genotype*, *age*, and *time* (follow-up time). As argued by Martinussen et al. [52], unmeasured confounding may arise between *Vitamin D status* and *mortality* due to behavioral and environmental factors. To estimate the causal effect of *Vitamin D status* on *mortality*, one may use the *filaggrin genotype* as instrumental variable, as reported by Martinussen et al. [52]. In our setup, the problem of interest is to verify that *filaggrin genotype* is a valid IV while *age* and *time* are not without the prior knowledge of causal structure.

Here, we also make comparisons with the sisVIVE algorithm and the IV-TETRAD algorithm. In the implementation, the significance level of all methods were set to 0.01. We have the following findings: (1) The output of IV-GIN is that *filaggrin genotype* is a valid IV while *age* and *time* are invalid, which indicates the effectiveness of our method. (2) The output of IV-TETRAD is an empty set. This is because there is only one valid IV, which violates the basic assumption (two or more variables are valid IVs in the system). (3) The output of sisVIVE is that *age* is a valid IV while *filaggrin genotype* and *time* are invalid. This implies that sisVIVE fails to find the valid IV, i.e., *filaggrin genotype*. One reason is that fewer than half of the variables are valid IVs in this dataset. These results again indicate that our algorithm has better performance than the other algorithms for selecting valid IVs.

## 9. Discussion

The preceding sections presented how to use IV-GIN conditions to select the set of candidate IVs relative a target causal influence 
X→Y
 from observed variables without prior knowledge of causal structure. In this section, we discuss the following two practical questions.

Is it possible to select IVs by learning the whole causal graph? In fact, it is challenging to discover the precise causal graph in the presence of arbitrary hidden variables. To show this fact, we apply the LRpSC+GES algorithm introduced by [43] to learn the diagrams of three scenarios in Section 7, respectively. For simplicity, we set sample size *N* = 5*k*. We identify the IVs according to the instrument criteria given the learned graph. In detail, if there is a direct edge between candidate variables *Z* and treatment *X* and there is no direct edge between candidate variables *Z* and outcome *Y*, we think variable *Z* is a candidate IV. (Note that this selection is relatively loose and not rigorous.) The results are given in the following Table 3. From the table, we can see that the *correct-selecting rate* is close to 0.1, which indicates that almost all valid IVs have been incorrectly removed from the candidate set of IVs. We note that the *selection commissions* are small in the three scenarios. The reason is that in most cases, a valid IV *Z* has a direct edge to both treatment *X* and outcome *Y* in the learned graph by LRpSC+GES algorithm. These findings show that given the learned graph by the LRpSC+GES algorithm, one can not correctly select the set of candidate IVs.

What happens if we have no background knowledge about 
X→Y
? Theoretically speaking, the IV-GIN algorithm does not need to restrict the relation between *X* and *Y*, and the output 
C
 of the IV-GIN algorithm contains all valid IVs for the ground-truth relation, e.g., 
X→Y
 or 
Y→X
. This is because we do not restrict the order of *X* and *Y* when we test whether 
({Z,W},{X,Y,W})
 satisfies the GIN condition in Theorem 2. To show this fact, for the three scenarios in Section 7, we reverse the order of *X* and *Y* to make it be 
Y→X
 and run our method in these graphs. For simplicity, we set sample size 
N=5k
. The results are shown in Table 4. From this table, we can see that two metrics are almost close to the original graph having the causal influence 
X→Y
 in Table 1, indicating that our method does not rule out the valid IVs relative to the ground-truth one relationship. It is noteworthy that if one needs to calculate the causal effect between *X* and *Y*, the causal order of *X* and *Y* must be given in advance. This is because the IV estimator is based on the order of *X* and *Y* (see Equation (Equation 1)).

## 10. Conclusions and Further Work

In this paper, we investigated the problem of testability of instrumental variables in linear non-Gaussian acyclic causal models. In particular, we proposed a necessary condition for detecting valid IVs relative to a target causal influence 
X→Y
, which is called the IV-GIN condition. We then gave the graphical implications of the IV-GIN condition in linear non-Gaussian acyclic causal models. We showed how the conditions of instrument criteria can be checked by exploiting the IV-GIN conditions. Moreover, we proposed a sequential method, which selected the set of candidate IVs for the target causal influence 
X→Y
 from the observational data without precise prior knowledge of causal structure.

The key difference from the existing research considering the testability of IV in a linear non-Gaussian acyclic causal model, such as IV-TETRAD [21,53], is that: (1) we studied the testability of both conditions 1 and 2 while IV-TETRAD only studies the testability of condition 2 (condition 1 as the prior knowledge), and that (2) we investigated the case where a single IV is present in the ground-truth graph while IV-TETRAD needs at least two IVs present. It is worth noting that one can verify the validity of condition 2a using the IV-GIN method in cases where at least two instruments are present in the ground-truth graph. However, the IV-TETRAD condition is too restrictive and rules out some valid IVs. Table 5 summarizes the testability results using the IV-GIN conditions and IV-TETRAD conditions.

There is another way of estimating the causal effect *X* on *Y* in a linear non-Gaussian acyclic causal model. For instance, Refs. [37,40] show that the causal effect between any two observed variables is partially identifiable (output the equivalence class of causal effects) by using overcomplete independent component analysis (O-ICA) [54]. One may naturally have the following question: is it necessary to select the IV for estimating the causal effect *X* on *Y*? In fact, as stated in [21], for O-ICA based methods, the size of the equivalence class of the identified causal effects could be very large, and the number of unmeasured confounders between *X* and *Y* is not clear. Therefore, it is necessary to select the valid IV relative to a target causal influence 
X→Y
 when there exist latent confounders between *X* and *Y* without prior knowledge of the number of latent confounders.

One direction of future work is to extend the IV-GIN condition to the case of a nonlinear additive noise model, and existing techniques [55,56,57] may help to address this issue.

## Figures and Tables

**Figure 1 entropy-24-00512-f001:**
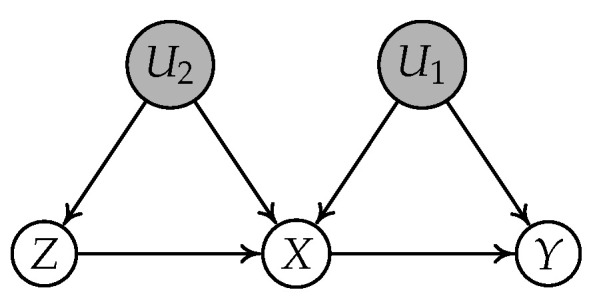
A simple instrumental variable example where *X* is treatment, *Y* is outcome, and *Z* is an IV relative to 
X→Y
.

**Figure 2 entropy-24-00512-f002:**
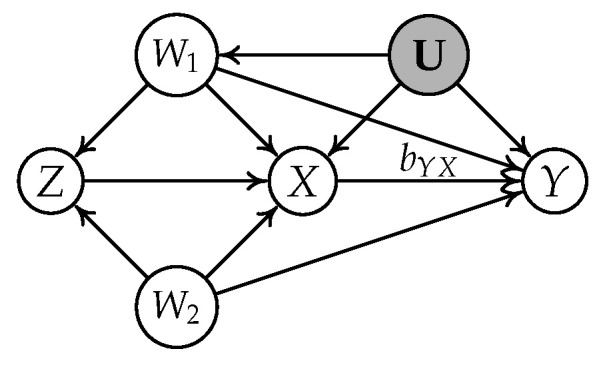
A typical instrumental variable model where *X* is treatment, *Y* is outcome, and *Z* is an IV conditioning on 
$\left\{W_{1},W_{2}\right\}$
 relative to 
X→Y
.

**Figure 3 entropy-24-00512-f003:**
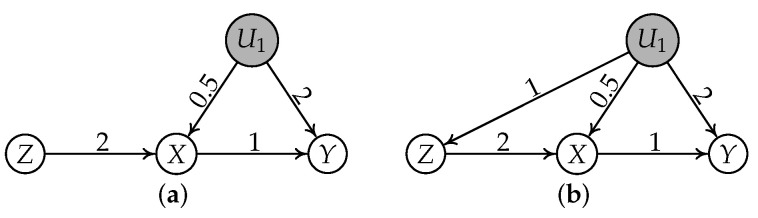
(**a**) *Z* is a valid IV for the relation 
X→Y
 and (**b**) *Z* is an invalid IV for the relation 
X→Y
.

**Figure 4 entropy-24-00512-f004:**
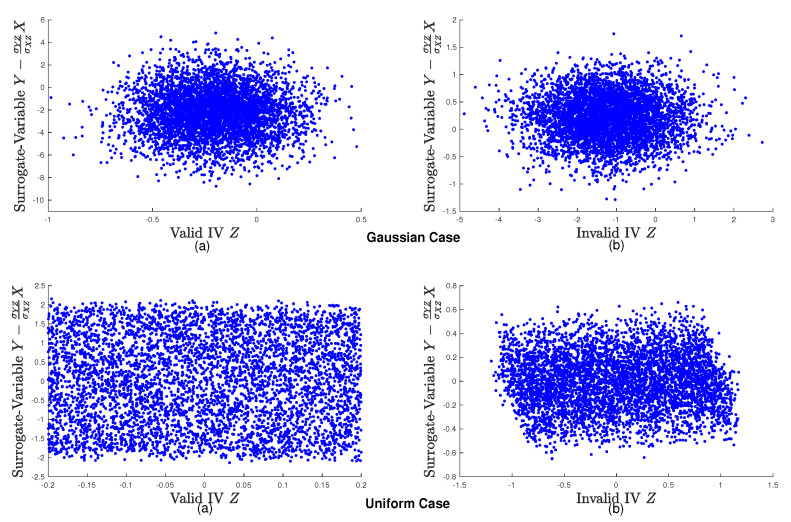
Illustration on the fact that non-Gaussianity leads to dependence between invalid IV *Z* and surrogate-variable 
$Y-\frac{\sigma_{YZ}}{\sigma_{XZ}}X$
. (**a**) Scatter plot of valid IV Z and surrogate-variable 
$Y-\frac{\sigma_{YZ}}{\sigma_{XZ}}X$
. (**b**) Scatter plot of invalid IV Z and surrogate-variable 
$Y-\frac{\sigma_{YZ}}{\sigma_{XZ}}X$
.

**Figure 5 entropy-24-00512-f005:**
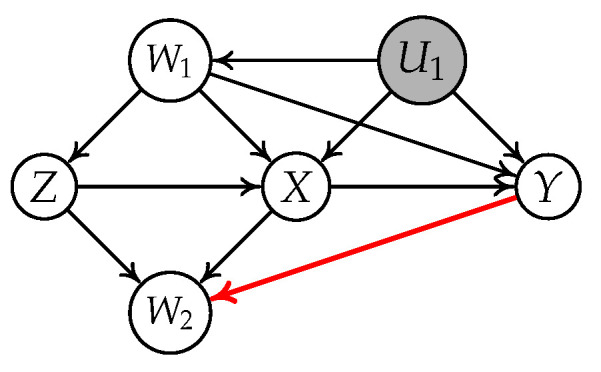
Causal graph where *Z* is a valid IV conditioning on 
$W_{1}$
 relative to 
X→Y
 but an invalid IV conditioning on 
$W_{2}$
 relative to 
X→Y
.

**Figure 6 entropy-24-00512-f006:**
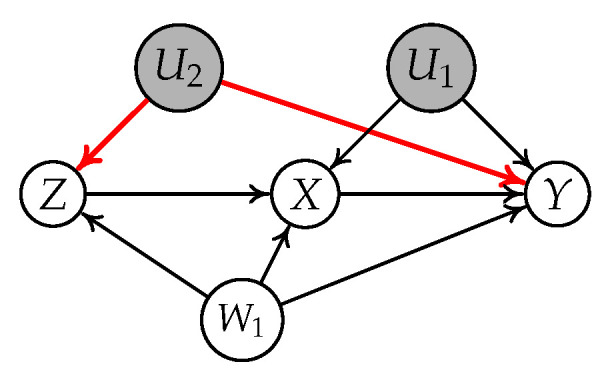
Causal graph where *Z* is an invalid IV conditioning on 
$W_{1}$
 relative to 
X→Y
 due to the nondirected path 
$Z\leftarrow U_{2}\to Y$
.

**Figure 7 entropy-24-00512-f007:**
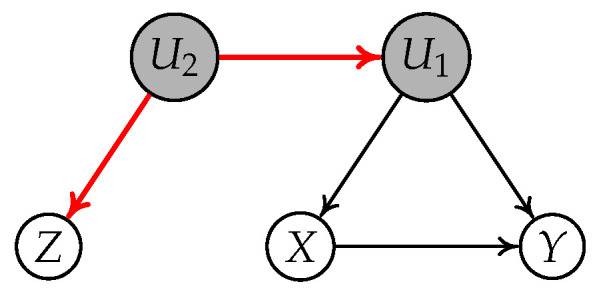
Causal graph where *Z* is a invalid IV conditioning on an empty set relative to 
X→Y
 but 
({Z},{Y,X})
 follows the GIN condition.

**Figure 8 entropy-24-00512-f008:**
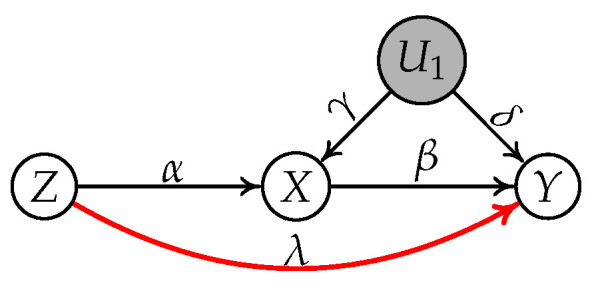
Causal graph where *Z* is an invalid IV conditioning on an empty set relative to 
X→Y
 due to the directed path 
Z→Y
.

**Figure 9 entropy-24-00512-f009:**
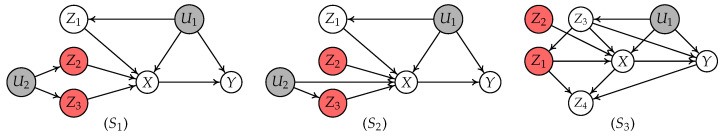
Three different scenarios used in our simulation studies.

**Table 1 entropy-24-00512-t001:** Performance of IV-GIN, sisVIVE, and IV-TETRAD on selecting valid IVs with different sample sizes.

		Correct-Selecting Rate ↑			Selection Commission ↓		
**Algorithm**		**IV-GIN (Ours)**	**sisVIVE**	**IV-TETRAD**	**IV-GIN (Ours)**	**sisVIVE**	**IV-TETRAD**
*Scenario $S_{1}$ *	1k	0.92	0.76	0.84	0.12	0.0	0.16
	3k	0.95	0.81	0.96	0.03	0.0	0.04
	5k	0.97	0.85	0.96	0.0	0.0	0.04
*Scenario $S_{2}$ *	1k	0.9	0.92	0.03	0.03	0.08	0.0
	3k	0.95	0.93	0.02	0.0	0.02	0.0
	5k	1.0	0.94	0.0	0.0	0.0	0.0
*Scenario $S_{3}$ *	1k	0.75	0.29	0.05	0.1	0.59	0.1
	3k	0.86	0.2	0.02	0.05	0.7	0.05
	5k	0.93	0.24	0.02	0.02	0.63	0.0

Note: ↑ means a higher value is better and ↓ means a lower value is better.

**Table 2 entropy-24-00512-t002:** Performance of IV-GIN, sisVIVE, and IV-TETRAD on selecting valid IVs with different effect of unmeasured confounders between treatment and outcome.

		Correct-Selecting Rate ↑			Selection Commission ↓		
**Algorithm**		**IV-GIN (Ours)**	**sisVIVE**	**IV-TETRAD**	**IV-GIN (Ours)**	**sisVIVE**	**IV-TETRAD**
*Scenario $S_{1}$ *	λ=0.125	0.96	0.83	0.92	0.06	0.01	0.08
	λ=0.25	0.85	0.72	0.86	0.01	0.0	0.01
*Scenario $S_{2}$ *	λ=0.125	0.98	0.93	0.02	0.04	0.06	0.0
	λ=0.25	0.92	0.91	0.0	0.08	0.1	0.0
*Scenario $S_{3}$ *	λ=0.125	0.89	0.22	0.05	0.03	0.58	0.02
	λ=0.25	0.85	0.2	0.03	0.07	0.61	0.0

Note: ↑ means a higher value is better and ↓ means a lower value is better.

**Table 3 entropy-24-00512-t003:** Performance of LRpSC+GES on selecting valid IVs with 5k sample sizes.

Metrics	Scenario $S_{1}$	Scenario $S_{2}$	Scenario $S_{3}$
Correct-selecting rate ↑	0.1	0.1	0.09
Selection commission ↓	0.0	0.12	0.3

**Table 4 entropy-24-00512-t004:** Performance of IV-GIN on selecting valid IVs with 5k sample sizes where the locations of nodes *X* and *Y* are swapped.

Metrics	Scenario $S_{1}$	Scenario $S_{2}$	Scenario $S_{3}$
Correct-selecting rate ↑	0.96	1.0	0.92
Selection commission ↓	0.01	0.0	0.04

**Table 5 entropy-24-00512-t005:** Summary of the testability results using the IV-GIN conditions presented in our paper and IV-TETRAD conditions presented in [21].

	Testability of Instrument Criteria		
**Method**	**Scenario $S_{1}$ **	**Scenario $S_{1}$ **	**Scenario $S_{1}$ **
IV-GIN (ours)	Fully	Partially	None
IV-TETRAD	None	Fully	None

## Data Availability

The simulated data can be regenerated using the codes, which can be provided to the interested user via an email request to the correspondence author. The Vitamin D Data used in the experiments come from the ivtools package of CRAN, which can be downloaded from https://mirrors.sjtug.sjtu.edu.cn/cran/web/packages/ivtools/index.html (accessed on 20 January 2022).

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
