# Peer review of "Testability of Instrumental Variables in Linear Non-Gaussian Acyclic Causal Models"

_entropy, 2022, doi:10.3390/e24040512_

Round 1
Reviewer 1 Report
The authors analyse the problem of selecting instrumental variables relative to a target causal influence X → Y from observational data generated by linear non-Gaussian acyclic casual models in the presence of unmeasured confounders. The authors propose a neccessary condition for finding a variables which can not serve as an instrument. The authors further develop a method to select the set of candidate instrumental variables given observational data. Given that finding a right set of instrument is very crucial in obtaining consistent estimates, the topic is important and will add significant contribution to the literature.
I have a specific comment about the nature of the causal relationship given instrument set. If the authors use the identified instruments (to examine the true causal influence X → Y) for examining the reverse causality, will it lead to a spurious causal inference from Y to X. The author could demonstrate the spurious on non-spurious nature of the results through the simulation execise.
Reviewer 2 Report
This paper consider the problem of selecting causal influences from observational data generated by linear non-Gaussian acyclic 2 casual models.
1) One main concern regards the literature review. Indeed, this problem is strictly connected with nonparametric approaches to network identification having sparse low rank structure where the sparse part aims to learn the causal influences while the low rank part aims to model the noise part.
2) The authors should also review the inference methods regarding the idetification of graphical models (e.g. sparse plus low rank, reciprocal structure) and made clear the modeling differences
3) The authors should also compare their algorithm with methods like in ones in 1). Indeed, one would really see that the whether the fact that the Gaussianity assumption is a real limit in the considered examples.
Round 2
Reviewer 2 Report
I am fine with the revised version.